# Conversion of Glycosylated Platycoside E to Deapiose-Xylosylated Platycodin D by Cytolase PCL5

**DOI:** 10.3390/ijms21041207

**Published:** 2020-02-11

**Authors:** Kyung-Chul Shin, Dae Wook Kim, Hyun Sim Woo, Deok-Kun Oh, Yeong-Su Kim

**Affiliations:** 1Research Institute of Bioactive-Metabolome Network, Konkuk University, Seoul 05029, Korea; hidex2@naver.com (K.-C.S.); deokkun@konkuk.ac.kr (D.-K.O.); 2Forest Plant Industry Department, Baekdudaegan National Arboretum, Bonghwa 36209, Korea; dwking@bdna.or.kr (D.W.K.); whs0428@bdna.or.kr (H.S.W.); 3Department of Bioscience and Biotechnology, Konkuk University, Seoul 05029, Korea

**Keywords:** *Platycodon grandiflorum*, platycoside, cytolase, deapiosylation, dexylosylation

## Abstract

Platycosides, the saponins abundant in Platycodi radix (the root of *Platycodon grandiflorum*), have diverse pharmacological activities and have been used as food supplements. Since deglycosylated saponins exhibit higher biological activity than glycosylated saponins, efforts are on to enzymatically convert glycosylated platycosides to deglycosylated platycosides; however, the lack of diversity and specificities of these enzymes has limited the kinds of platycosides that can be deglycosylated. In the present study, we examined the enzymatic conversion of platycosides and showed that Cytolase PCL5 completely converted platycoside E and polygalacin D3 into deapiose-xylosylated platycodin D and deapiose-xylosylated polygalacin D, respectively, which were identified by LC-MS analysis. The platycoside substrates were hydrolyzed through the following novel hydrolytic pathways: platycoside E → platycodin D3 → platycodin D → deapiosylated platycodin D → deapiose-xylosylated platycodin D; and polygalacin D3 → polygalacin D → deapiosylated polygalacin D → deapiose-xylosylated polygalacin D. Our results show that cytolast PCL5 may have a potential role in the development of biologically active platycosides that may be used for their diverse pharmacological activities.

## 1. Introduction

Platycosides are pharmaceutically active saponins present in the root of *Platycodon grandiflorum*, also known as balloon flower, and have been used as a part of the medicinal diet in Northeast Asia [1,2]. Their pharmacological properties include anti-oxidant [3,4], anti-inflammatory [5,6], anti-tumor [7,8,9], anti-allergy [10,11], anti-obesity [12,13,14], and immune-modulatory effects [15,16,17].

Platycosides are composed of a triterpenoid aglycone and two sugar chains at the C-3 and C-28 sites. Most of the aglycones are platycodigenin-type (CH_2_OH at C-24) and polygalacic acid-type (CH_3_ at C-24) accounting for about 50% and 30% of the total platycosides, respectively [18,19] (Figure 1).

Deglycosylation of saponins increases their biological activity as it enhances their bioavailability and cell permeability due to reduced size [20,21]. Methods such as heating [22], acid treatment [23], fermentation [24], whole-cell reaction [25], and enzymatic conversion [26] have been reported to hydrolyze the sugar moieties in glycosylated saponins. Among these methods, enzymatic conversion showed the highest yield, efficiency, and specificity. Encouraged by these results, various enzymes were used to deglycosylate platycosides. The β-glycosidases from *Caldicellulosiruptor bescii* [27], *Caldicellulosiruptor owensensis* [28], and *Aspergillus usamii* [29], snailase [30], laminarinase [31], and cellulase [32] converted platycoside E to platycodin D via platycodin D3 as the intermediate. On the other hand, the one from *Dictyoglomus turgidum* deglycosylated platycoside E to deglucosylated platycodin D via platycodin D3 and platycodin D as intermediates [33]. Most of these enzymes hydrolyzed the glucose molecules at the C-3 position of platycosides. However, when the crude enzyme from *Aspergillus niger* was reacted with platycodin D, apiose and xylose moieties were removed from C-28 [34]. Nevertheless, deapiose-xylosylated platycodin D has never been produced quantitatively and no other type of platycoside has been used to show deapiosylation and dexylosylation.

Cytolase PCL5, an acid-resistant pectinase from *Aspergillus niger*, has been used to hydrolyze glycosides and flavonoids [35,36,37,38]. Cytolase PCL5 had higher activity on ginsenosides than other commercial enzymes [36,38], therefore it was newly applied to the conversion of platycosides in the present study.

In this study, we investigated the enzymatic conversion of platycosides, and to this end, we used Cytolase PCL5 to convert platycoside E and polygalacic acid D3, the major components present in Platycodi radix, and the chemical structures of deapiose-xylosylated platycodin D and deapiose-xylosylated polygalacic acid D were identified. Further, these platycosides were quantitated to analyze reaction efficiency.

## 2. Results and Discussion

### 2.1. Identification of Products Obtained by the Action of Cytolase PCL5 on Platycoside E and Polygalacin D3 by

Cytolase PCL5 was used in the present study to catalyze the bioconversion of platycoside E and polygalacin D3. The products were analyzed by HPLC using a reversed-phase hydrosphere C18 column and reference standards including platycoside E, platycodin D3, platycodin D, deapiosylated platycodin D, polygalacin D3, and polygalacin D (Figure 2). Our results showed that the enzyme converted platycoside E to a product with the retention time the same as deapiosylated platycodin D (26.3 min), whereas the retention time of another product was 25.7 min. Polygalacin D3 was catalyzed into two unknown products with retention times (2’, 27.0 min and 3’, 26.7 min) that didn’t match the standards.

The three unknown products were characterized using the LC/TOF-MS in the positive and negative mode. Their theoretical mass, molecular formula, and observed mass are summarized in Table 1. The total ion chromatogram (TIC) of the three unknown products is shown in Appendix A. For compound **1**′ in ESI+, [M+H]^+^ pseudomolecular ions presented at m/z 961.4966, which corresponded to the molecular formula of C_47_H_76_O_20_. In ESI-, the [M+HCOO]^−^ adduct was detected in compound **2**′ and **3**′. Due to high resolution, compound identification was based on the exact mass of each molecule. The mass accuracy of the experimental mass data compere with the theoretical value was less than 5 ppm. According to HRMS (High-Resolution Mass Spectrometry) data, three unknown products were determined to be deapiosylated platycodin D (**1**′), deapiosylated polygalacin D (**3**′), and deapiosylated platycodin D (**2**′), respectively.

### 2.2. Effects of pH and Temperature on the Hydrolytic Activity of Cytolase PCL5

Optimum conditions for Cytolase PCL5 activity have been reported in a wide range of pH (2.5–5.0) and temperature (10–55 °C) [37]. In this study, we analyzed the hydrolytic activity of Cytolase PCL5 at 40–65 °C over pH 4.0–7.0 using platycoside E as a substrate. We observed the maximum activity at pH 5.0 (Figure 3a). Interestingly, the enzyme activity at pH 4.0 was less than 25% of the maximum activity, indicating that it was sensitive to pH change when platycoside E was used as the substrate. Next, we investigated the effect of temperature on the enzyme activity and our results showed maximum activity at 50 °C (Figure 3b). Further, the optimal pH and temperature of Cytolase PCL5 for flavanone rutinoside [35] (pH 4.0 and 60 °C), white ginseng extract [36] (pH 4.3 and 55 °C), and red ginseng powder [38] (pH 5.0 and 45 °C) were similar regardless of the glycosides were as the substrate.

### 2.3. Substrate Specificity of Cytolase PCL5 for Platycosides

The substrate specificity of Cytolase PCL5 was investigated using platycodigenin- and polygalacic acid-type platycosides as substrates (Table 2). The specific activity of the enzyme was observed in the following order: playtcoside E > platycodin D3 > platycodin D > deapiosylated platycodin D, and polyglacin D3 > polygalacin D > deapiosylated polygalacin D. These results indicated that Cytolase PCL5 acted efficiently at outer glucose moiety (C-3) than glycosides at C-28 and apiose than xylose at C-28. The hydrolysis of glucose at C-3 and apiose at C-28 were 3.1- and 1.3-fold higher in polygalacic acid-type than in platycodigenin-type platycosides. In contrast, the hydrolysis of xylose at C-28 was 1.2-fold higher in platycodigenin-type than polygalacic acid-type. 

Based on the results of substrate specificity, biotransformation pathways of platycoside E by Cytolase PCL5 were determined as follows: platycoside E → platycodin D3 → platycodin D → deapiosylated platycodin D → deapiose-xylosylated platycodin D (Figure 4). The biotransformation pathways of polygalacin D3 were similar to the sequence of transformations of platycodin D3 as follows: polygalacin D3 → polygalacin D → deapiosylated polygalacin D → deapiose-xylosylated polygalacin D. To the best of our knowledge, the pathways of platycoside E to deapiose-xylosylated platycodin D and polygalacin D3 to deapiose-xylosylated polygalacin D have not been reported so far, and the present study is the first report on deapiosylated polygalacin D and deapiose-xylosylated polygalacin D.

### 2.4. Biotransformations of Platycoside E and Polygalacin D3 into Deapiose-Xylosylated Platycosides by Cytolase PCL5

The biotransformations of platycoside E and polygalacin D3 to deapiose-xylosylated platycodin D and deapiose-xylosylated polygalacin D, respectively, were performed with 0.5 mg/mL enzyme and 1 mg/mL platycoside as the substrate. The enzyme converted platycoside E and polygalacin D3 to deapiose-xylosylated platycodin D and deapiose-xylosylated polygalacin D within 12 and 15 h, respectively, with concentrations of 0.62 and 0.69 g/L and productivities of 51.70 and 45.95 mg/L/h, respectively (Figure 5). The concentration of deapiose-xylosylated platycosides were lower than that of substrate, because the sugar moieties of the substrates were removed from the substrate. Platycosides such as platycoside E, platycodin D3, and polygalacin D3 were not detected after 5 min due to their rapid conversion. Using food-grade commercial enzyme for the production of minor platycosides is economically advantageous due to high enzyme activity and stability. Furthermore, the enzyme can be used as an industrial biocatalyst through further processes such as immobilization with enhancements in stability and biocatalyst functionality [39,40,41,42].

## 3. Materials and Methods

### 3.1. Materials

Cytolase PCL5 was purchased from DSM Food Specialties (Heerlen, Netherlands). The platycoside standards (platycoside E, platycodin D3, platycodin D, deapiosylated platycodin D, and polygalacin D3) were purchased from Ambo Laboratories (Daejeon, Republic of Korea). Polygalacin D was kindly provided by Dr. Dae Young Lee of the National Institute of Horticultural and Herbal Science, Eumseong, Republic of Korea. Deapiose-xylosylated platycodin D, deapiosylated polygalacin D, and deapiose-xylosylated polygalacin D were purified from the reactants that underwent catalysis by Cytolase PCL5. Platycosides were prepared by preparative high-performance liquid chromatography (Prep-HPLC) (Agilent 1260; Agilent, Santa Clara, CA, USA) using a Hydrosphere C18 prep column (10 × 250 mm, 5 μm particle size; YMC, Kyoto, Japan). The column was eluted with water at a flow rate of 4.7 mL/min at 30 °C and the products were detected by measuring the absorbance at 203 nm. The prepared platycosides were used as standards for quantitative and qualitative analysis.

### 3.2. Hydrolytic Activity Assay

Unless stated otherwise, the reactions were performed at 50 °C for 10 min in 50 mM citrate/phosphate buffer (pH 5.0) containing 0.05 mg/mL Cytolase PCL5 and 0.2 mg/mL platycoside. The specific activities of Cytolase PCL5 for platycoside E, platycodin D3, platycodin D, deapiosylated platycodin D, deapiose-xylosylated platycodin D, polygalacin D3, polygalacin D, deapiosylated polygalacin D, and deapiose-xylosylated polygalacin D were evaluated at various concentrations of the enzyme (0.005–0.5 mg/mL) and 0.4 mg/mL of each platycoside for 10 min at 50 °C and pH 5.0. The effects of pH and temperature on the activity of Cytolase PCL5 were examined by varying the pH from 4.0 to 7.5 at 50 °C, and by varying the temperature from 40 to 65 °C at a pH of 5.0, respectively.

### 3.3. Biotransformation

The biotransformations of platycoside E into deapiose-xylosylated platycodin D, and polygalacin D3 into deapiose-xylosylated polygalacin D, were performed for 15 and 18 h at 50 °C in 50 mM citrate/phosphate buffer (pH 5.0) containing 0.5 mg/mL Cytolase PCL5 and 1 mg/mL platycoside E and polygalacin D3, respectively.

### 3.4. HPLC Analysis

An equal amount of n-Butanol was added to the reaction mixture to stop the reaction and extract the platycosides. This resulted in the separation of reactants into aqueous and n-butanol soluble fractions. The n-butanol soluble fraction of the extract was dried to evaporate the butanol completely. Subsequently, dried residues containing platycosides were dissolved in methanol and analyzed using an HPLC system (Agilent 1100) at 203 nm with a hydrosphere C18 column (4.6 × 150 mm, 5 μm particle size; YMC, Kyoto, Japan). The column was eluted at 30 °C with a gradient of solvent A (acetonitrile) and solvent B (water) from 10:90 to 40:60 for 30 min, from 40:60 to 90:10 for 15 min, from 90:10 to 10:90 for 5 min, and constant at 10:90 for 10 min at a flow rate of 1 mL/min. The linear calibration curves relating the logarithmic value of the peak areas to the concentrations of platycosides were constructed using the standard solutions of platycosides (0.2 to 1.0 mM) and the curves were used to determine the platycosides concentrations.

### 3.5. Liquid Chromatography-Mass Spectrometry Analysis of Platycosides

The platycosides were further analyzed using a Waters SYNAPT G2-Si HDMS instrument (Waters Co. Taunton, MD, USA). This system comprises a Waters Acquity UPLC system coupled to a quadrupole time-of-flight mass spectrometer. The samples were eluted using a Waters Acquity BEH C18 column (100 mm × 2.1 mm, 1.7 mm) set at 40 °C. All solvents used were LC-MS grade and ultra-pure 18.2 MΩ water was used at each step. Mobile phase A consisted of water + 0.1% formic acid while mobile phase B was acetonitrile + 0.1% formic acid. The gradients elution was 100% A (0.00 min) 95% A+ 5% B (0.00–1.00 min), 5% A+ 95% B (1.00–9.00 min), 100% B (9.00–10.50 min), 100% A (10.50–11.00 min), and 100% A (11.00–12.50 min) with the flow rate of 0.4 mL/min and injection volume of 3 μL for positive mode and 2 μL for negative mode.

Mass spectrometry detection was conducted through electrospray ionization using an MS^E^ (Mass Spectrometry ^Elevated Energy^) centroid experiment performed in both positive and negative modes and screened the m/z scan range of 50–1500 Da with the analyzer set to resolution mode at FWHM (Full With at Half Maximum). Scanning was performed every 0.5 s. The collision energy was set for two functions: function one at low energy with no collision energy applied and function two at high energy using voltage 2.00 kV, sampling cone 10 V, source temperature 100 °C, desolvation temperature 400 °C, and desolvation gas flow 700 L/h. The accurate mass was initially calibrated prior to sample analysis by direct infusion of sodium iodide calibrant solution. In addition, leucine encephalin lock mass solution (2 ng/ μL) was infused at 5 μL/min, parallel to the mobile phase flow, scanned and automatically corrected to verify exact mass, thereby ensuring high mass accuracy (<5 ppm) throughout the scan range over the course of the submitted sequence. Masslynx v.41 software was used to control the instrument and perform data analysis.

## 4. Conclusions

In this study, we showed that Cytolase PCL5 indicated new substrate specificity by hydrolyzing not only the outer glucose moieties at C-3 of platycoside E or polygalacin D3, respectively but also xylose and apiose at C-28. Cyotolase PCL5 is an enzyme blend rather than an individual enzyme, therefore it may be specific for a variety of sugars. The chemical structures of the unknown products obtained by the platycosides catalysis were identified by LC-MS, and the deapiose-xylosylated platycodin D and deapiose-xylosylated polygalacin D were identified as final products of the reaction. Cytolase PCL5 completely converted the platycoside substrates to the final products without further hydrolysis, indicating that the enzyme was effective for the production of deapiose-xylosylated platycodin D and deapiose-xylosylated polygalacin D. The food-grade commercial enzyme is economically feasible for industrial application through further immobilization technology. To the best of our knowledge, this is the first study to report the quantitative production of deapiose-xylosylated platycodin D and deapiose-xylosylated polygalacin D. One of the future developments of the process may be the sue of immobilize enzymes, that way the enzyme recovering will be simplified, and also some improvements on enzyme stability (that can permit to broaden the range of uses of the enzyme) and even improve specificity, selectivity, or activity of the enzyme. It can also reduce inhibitions or permit the one step purification-immobilization of the enzyme.

## Figures and Tables

**Figure 1 ijms-21-01207-f001:**
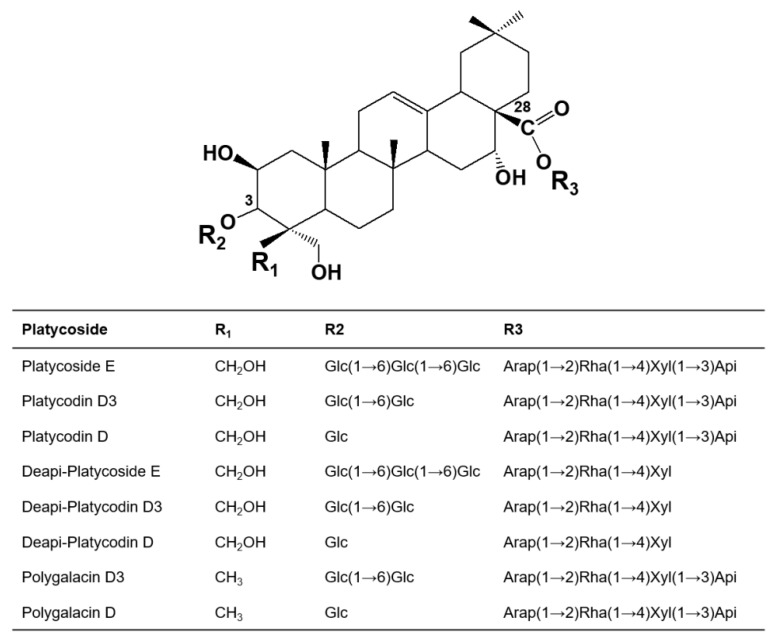
Chemical structures of platycodigenin- and polygalacic acid-type platycosides found in the roots of *Platycodon grandiflorum*. Platycosides contain glycosides at C-3 and C-28 positions. The glycosides at C-3 are Glc, Glc-Glc, and Glc-Glc-Glc. The glycosides at C-28 are Arap-Rha-Xyl and Arap-Rha-Xyl-Api. Glc, β-d-glucopyranosyl-; Arap, α-l-arabinopyranosyl-; Rha, α-l-rhamnopyranosyl-; Xyl, β-d-xylopyranosyl-; and Api, β-d-apiofuranosyl-.

**Figure 2 ijms-21-01207-f002:**
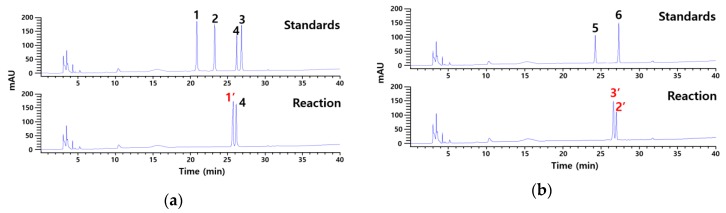
HPLC profiles showing the unknown products obtained after the catalysis of (**a**) platycoside E and (**b**) polygalacin D3 by Cytolase PCL5. Standards used for analysis included platycoside E, platycodin D3, platycodin D, deapiosylated platycodin D, polygalacin D3, and polygalacin D (represented with 1, 2, 3, 4, 5, and 6). Red colored 1’, 2’, and 3’ represent unknown products from platycoside E and polygalacin D3, respectively.

**Figure 3 ijms-21-01207-f003:**
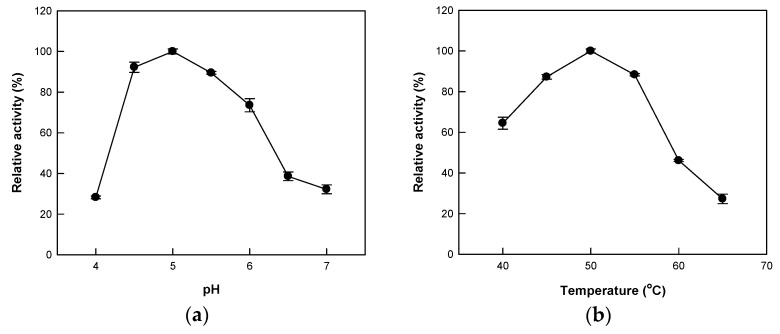
The effect of temperature and pH on the activity of Cytolase PCL5 on platycoside E. (**a**) Effect of pH. The reactions were carried out with 0.4 mM platycoside E in 50 mM citrate/phosphate buffer (pH 4.0–7.0) at 50 °C for 10 min. (**b**) Effect of temperature. The reactions were carried out with 0.4 mM platycoside E in 50 mM citrate/phosphate buffer (pH 5.0) at 40–65 °C for 10 min. Data are represented as the means of triplicate experiments, and error bars represent the standard deviation.

**Figure 4 ijms-21-01207-f004:**
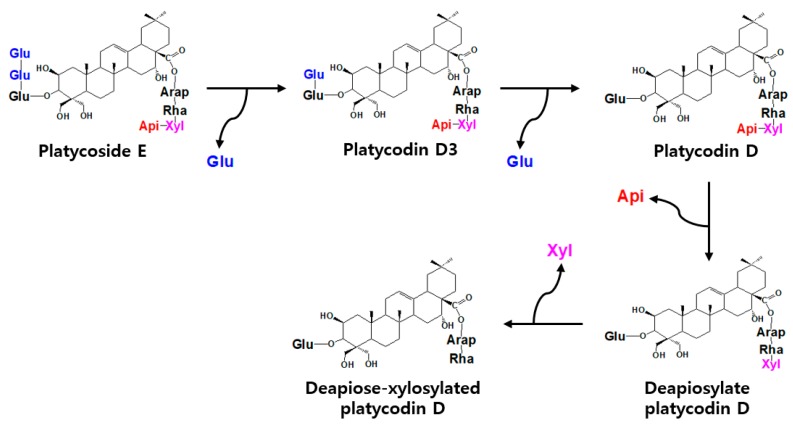
Biotransformation of platycoside E into deapiose-xylosylated platycodin D via platycodin D3, platycodin D, and deapiosylated platycodin D.

**Figure 5 ijms-21-01207-f005:**
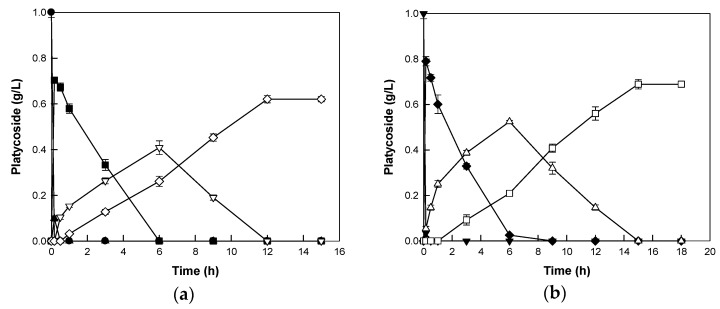
Biotransformations of platycodigenin- and polygalacic acid-type platycosides by Cytolase PCL5. (**a**) Biotransformation of platycoside E (closed circles) into deapiose-xylosylated platycodin D (open diamonds) via platycodin D3 (closed triangles), platycodin D (closed squares), and deapiosylated platycodin D (open inverted triangles). (**b**) Biotransformation of polygalacin D3 (closed inverted triangles) into deapiose-xylosylated polygalacin D (open squares) via polygalacin D (closed diamonds) and deapiosylated polygalacin D (open triangles). Data are represented as the means of experiments done in triplicates, and error bars represent the standard deviation.

**Table 1 ijms-21-01207-t001:** Liquid chromatography quadrupole time-of-flight mass spectrometry (LC-QTOF/MS) of unknown compounds.

Compound	Formula	Selected Ion	*m/z* _experimetnal_	*m/z* _calculated_	Error	
					mDa	ppm
Deapiose-xylosylated platycodin D (**1**′)	C_47_H_76_O_20_	[M+H]^+^	961.4966	961.5003	−3.7	3.8
Deapiose-xylosylated polygalacin D (**2**′)	C_48_H_77_O_21_	[M–H + HCO_2_H]^−^	989.4985	989.4957	2.8	2.8
Deapiosylated polygalacin D (**3**′)	C_53_H_85_O_25_	[M–H + HCO_2_H]^−^	1121.5436	1124.5380	5.6	5.0

**Table 2 ijms-21-01207-t002:** Substrate specificity of Cytolase PCL5 for platycosides.

Substrate	Product	Specific Activity (nmol/min/mg)
PE	PD3	15481.2 ± 27.5
PD3	PD	270.9 ± 11.0
PD	Deapi-PD	30.8 ± 3.2
Deapi-PD	Deapi-Dexyl-PD	13.3 ± 1.5
Deapi-xyl-PD	−	ND
PGD3	PGD	844.4 ± 10.2
PGD	Deapi-PGD	38.6 ± 1.6
Deapi-PGD	Deapi-Dexyl-PGD	10.7 ± 0.5
Deapi-xyl-PGD	−	ND

PE, platycoside E; PD3, platycodin D3; PD, platycodin D; PGD3, polygalacin D3; PGD, polygalacin D; Deapi, deapiosylated; Deapi-xyl, deapiose-xylosylated; ND, specific activity not detected by the analytical methods used in this study; −, no enzyme reaction product by the analytical methods used in this study.

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
