# Peer review of "Conversion of Glycosylated Platycoside E to Deapiose-Xylosylated Platycodin D by Cytolase PCL5"

_ijms, 2020, doi:10.3390/ijms21041207_

Round 1

Reviewer 1 Report

In this manuscript the authors describe the use of a commercial enzyme blend, Cytolase PCL5 to deglycosylate the saponins, platycoside E and polygalactin D3. Prior studies have used individual enzymes to hydrolyze glucose from platycoside E, or a crude enzyme preparation from Aspergillus niger to demonstrate that removal of apiose and xylose are possible at carbon 28. Here, the authors selected Cytolase PCL5, a fungal commercial enzyme preparation to treat platycoside E or polygalactin D3 with. The authors used HPLC, and LC-MS to identify the end products of those reactions. Overall, they were able to observe the sequential deglycosylation of platycosides to their deapiosylated and dexylosidated forms.

Throughout the manuscript, Cytolase PCL5 itself and the activity of this enzyme mixture is referred to as an enzyme, rather than a commercial enzyme mixture. “Cytolase” is even listed as a keyword on line 25. The fact that multiple enzymes are responsible for the activity observed needs to be described more clearly in the manuscript.

Minor Comments

Page 1 Line 40: beta-glucosidases shouldn't be capitalized

Page 1 Line 42: Specify in this sentence which enzyme from Dictyoglomus turgidum is being referred to.

Page 1 Line 43: The description of the D. turgidum enzyme hydrolysis of platycoside E to platycodin D is no different than that described in like 42. Also, did the authors mean to list platycodin D as an intermediate if it's a final product?

Page 2 Line 62: This background information on the Cytolase PCL5 enzyme blend would be more useful in the Introduction, along with more of an explanation on why the authors selected this enzyme mixture.

Page 3 Lines 89 - 90: Please provide citations for the past studies on optimal conditions for cytolase PCL5

Page 5 Line 125, and elsewhere: Cytolase should be capitalized since it is a commercial enzyme blend.

page 7 line 216: Cytolase PCL5 is an enzyme blend, not an individual enzyme. The discussion needs to reflect this.

Author Response

Response to Reviewer #1

Page 1 Line 40: beta-glucosidases shouldn't be capitalized

Response) Thank you for your pointing out. As you mentioned, ‘β-Glucosidases’ was revised to ‘β-glucosidases’. (Line 39 of the revised manuscript)

Page 1 Line 42: Specify in this sentence which enzyme from Dictyoglomus turgidum is being referred to. Page 1 Line 43: The description of the D. turgidum enzyme hydrolysis of platycoside E to platycodin D is no different than that described in like 42. Also, did the authors mean to list platycodin D as an intermediate if it's a final product?

Response) Thank you for your good comment. We mistyped ‘deglucosylated platycodin D’ as ‘platycodin D’ in the original manuscript. Thus, we revised the sentence to ‘On the other hand, the one from Dictyoglomus turgidum deglycosylated platycoside E to deglucosylated platycodin D via platycodin D3 and platycodin D as intermediates’ in the revised manuscript. (Line 41-43 of the revised manuscript)

Page 2 Line 62: This background information on the Cytolase PCL5 enzyme blend would be more useful in the Introduction, along with more of an explanation on why the authors selected this enzyme mixture.

Response) Thank you for your suggestion. As you suggested, the background information on the Cytolase PCL5 was moved to Introduction part and the reason why the enzyme was chosen was newly described. (Line 48-51of the revised manuscript)

Page 3 Lines 89 - 90: Please provide citations for the past studies on optimal conditions for cytolase PCL5.

Response) Thank you for your pointing out. As you suggested, citation was newly added. (Line 92 of the revised manuscript)

Page 5 Line 125, and elsewhere: Cytolase should be capitalized since it is a commercial enzyme blend.

Response) Thank you for your pointing out. As you mentioned, all of ‘Cytolase’ was capitalized in the revised manuscript.

page 7 line 216: Cytolase PCL5 is an enzyme blend, not an individual enzyme. The discussion needs to reflect this.

Response) Thank you for your suggestion. As you suggested, related sentence was newly added in Conclusions part. (Line 214-215 of the revised manuscript)

Reviewer 2 Report

This work is a continuation of research on the sequence of Platycosides deglycosylation. In previous publications, autors described the enzymatic deglycosylation at C-3 carbon of platycosides. In the current publication, as a result of using PCL5 cytolase, they also described deglycosylation at  C-28 carbon. This research was inspired by the publication: Wie, H. J.; Zhao, H.L.; Chang, J.H.; Kim, Y.S.; Hwang, I.K.; Ji, G.E. Enzymatic Modification of Saponins from Platycodon Grandiflorum with Aspergillus Niger. J. Agric. Food Chem. 2007, 55, 8908-13.

I have thoroughly analyzed the manuscript, and I think it should be published in Int. J. Mol. Sci. after minor corrections.

Authors should definitely rethink the title of this work and replace it with a title more accurately describing the subject of the paper.

Figure 4 is not clear. Please divide it into two separate schemes or delete scheme b and fully describe in the text that biotransformation course of Polygalacin D3 are similar to the sequence of transformations of Platycodin D3.

The biotransformation course was determined on the basis of data obtained from the MS analysis. I regret that the authors did not isolate described compounds and confirm their structures by X-ray analysis or NMR spectroscopy. However, due to the specificity of the tested compounds, such isolation would require the authors to be much more involved in exploring the separation methods. Of course, this article is a compact and complete whole, but I suggest that such attempts will be made in the future and the spectral data of these compounds will be published in subsequent papers.

Author Response

Response to Reviewer #2

Authors should definitely rethink the title of this work and replace it with a title more accurately describing the subject of the paper.

Response) Thank you for your suggestion. As you suggested, we revised the title as ‘Conversion of glycosylated platycoside E to deapiose-xylosylated platycodin D by Cytolase PCL5’.

Figure 4 is not clear. Please divide it into two separate schemes or delete scheme b and fully describe in the text that biotransformation course of Polygalacin D3 are similar to the sequence of transformations of Platycodin D3.

Response) Thank you for your suggestion. As you suggested, we deleted scheme b in Figure 4 and revised the related sentences in the revised manuscript. (Line 119-124 and 128-130 of the revised manuscript)

3. The biotransformation course was determined on the basis of data obtained from the MS analysis. I regret that the authors did not isolate described compounds and confirm their structures by X-ray analysis or NMR spectroscopy. However, due to the specificity of the tested compounds, such isolation would require the authors to be much more involved in exploring the separation methods. Of course, this article is a compact and complete whole, but I suggest that such attempts will be made in the future and the spectral data of these compounds will be published in subsequent papers.

Response) Thank you for your concern. As you mentioned, we determined the platycosides only on the basis of data obtained from the MS analysis because we judged that it was sufficient based on the deglycosylation by the enzyme. As you suggested, we will perform the purification and NMR analysis of the compounds in subsequent papers for more accurate analysis.